# Hearing Impairment in a Mouse Model of Diabetes Is Associated with Mitochondrial Dysfunction, Synaptopathy, and Activation of the Intrinsic Apoptosis Pathway

**DOI:** 10.3390/ijms22168807

**Published:** 2021-08-16

**Authors:** Ah-Ra Lyu, Tae-Hwan Kim, Sun-Ae Shin, Eung-Hyub Kim, Yang Yu, Akanksha Gajbhiye, Hyuk-Chan Kwon, A Reum Je, Yang Hoon Huh, Min Jung Park, Yong-Ho Park

**Affiliations:** 1Department of Otolaryngology-Head and Neck Surgery, College of Medicine, Chungnam National University, Daejeon 35015, Korea; ahmilove@naver.com (A.-R.L.); ellision21@hanmail.net (E.-H.K.); ent-yuyang0924@naver.com (Y.Y.); 2Department of Medical Science, College of Medicine, Chungnam National University, Daejeon 35015, Korea; akanksha0323@gmail.com (A.G.); kwoneric@naver.com (H.-C.K.); 3Biomedical Convergence Research Center, Chungnam National University Hospital, Daejeon 35015, Korea; czkth@naver.com; 4Brain Research Institute, College of Medicine, Chungnam National University, Daejeon 35015, Korea; ttd0707@naver.com; 5Electron Microscopy Research Center, Korea Basic Science Institute, Cheongju 28119, Korea; areum83@kbsi.re.kr (A.R.J.); hyh1127@kbsi.re.kr (Y.H.H.)

**Keywords:** cochlear mitochondria, microangiopathy, synaptopathy, sensory loss, apoptosis, necroptosis

## Abstract

Although previous studies continuously report an increased risk of hearing loss in diabetes patients, the impact of the disease on the inner ear remains unexplored. Herein, we examine the pathophysiology of diabetes-associated hearing impairment and cochlear synaptopathy in a mouse model of diabetes. Male B6.BKS(D)-*Lepr*^db^/J (db/db, diabetes) and heterozygote (db/+, control) mice were assigned into each experimental group (control vs. diabetes) based on the genotype and tested for hearing sensitivity every week from 6 weeks of age. Each cochlea was collected for histological and biological assays at 14 weeks of age. The diabetic mice exerted impaired hearing and a reduction in cochlear blood flow and C-terminal-binding protein 2 (CtBP2, a presynaptic ribbon marker) expression. Ultrastructural images revealed severely damaged mitochondria from diabetic cochlea accompanied by a reduction in Cytochrome c oxidase subunit 4 (COX4) and CR6-interacting factor 1 (CRIF1). The diabetic mice presented significantly decreased levels of platelet endothelial cell adhesion molecule (PECAM-1), B-cell lymphoma 2 (BCL-2), and procaspase-9, but not procaspase-8. Importantly, significant changes were not found in necroptotic programmed cell death markers (receptor-interacting serine/threonine-protein kinase 1, RIPK1; RIPK3; and mixed lineage kinase domain-like pseudokinase, MLKL) between the groups. Taken together, diabetic hearing loss is accompanied by synaptopathy, microangiopathy, damage to the mitochondrial structure/function, and activation of the intrinsic apoptosis pathway. Our results imply that mitochondrial dysfunction is deeply involved in diabetic hearing loss, and further suggests the potential benefits of therapeutic strategies targeting mitochondria.

## 1. Introduction

Sudden or gradual hearing loss has been reported in patients with diabetes [1]. According to studies in the USA and Japan, the incidence of hearing loss in patients with diabetes is double that in the general population [2,3]. Over 422 million people worldwide have diabetes, and around 466 million people worldwide have disabling hearing loss. There appears to be a great deal of overlap between these two populations. Autopsy studies of diabetes patients suggest that this association is caused by neuropathy, including spiral ganglion atrophy, a reduction in the number of spiral lamina nerve fibers, the degeneration of the vestibulocochlear nerve myelin sheath, and the thickening of the capillary walls of the stria vascularis and small arteries [4,5,6,7].

Cochlear microcirculation plays an important role in cochlear physiology. Hyperglycemia and hyperlipidemia are associated with increased blood viscosity and circulation disorders. Inner ear diseases are often associated with microcirculation disorders, particularly those that involve the stria vascularis. The cell-surface protein, PECAM-1, is highly expressed on the surface of endothelial cells, specifically in cell–cell junctions, and plays important roles in the maintenance of vascular integrity, cell adhesion, and signaling. The 130 kD protein PECAM-1 functions as an adhesive stress response protein to maintain vascular integrity and signaling. Recent studies have indicated that PECAM-1/CD31 transmits survival signals into blood and vascular cells, where it functions as a potent inhibitor of mitochondria-dependent apoptosis [8,9].

Mitochondria are vital organelles responsible for generating the energy required for cell survival, the transduction of cellular signal cascades, and calcium (Ca^2+^) buffering. The citric acid cycle and fatty acid β-oxidation occurs in the mitochondrial matrix, and the respiratory chain subunits where oxidative phosphorylation occurs are embedded in the mitochondrial inner membrane. CR6-interacting factor-1, also known as GADD45-associated family protein, is a key molecule in the translation of mitochondrial oxidative phosphorylation subunits and their translocation into the mitochondrial inner membrane [10,11,12]. CRIF1 is an essential regulatory factor involved in mitochondrial ribosome-mediated synthesis and the insertion of oxidative phosphorylation polypeptides into the inner membranes of mitochondria [12]. Animal models of *Crif1* deficiency have revealed progressive beta cell failure (when deleted in pancreatic beta cells) [11] and insulin-resistance and inflammation (when deleted in adipose tissues) [13], indicating its role as a key regulator of mitochondrial function and metabolism. Decreased levels of this protein are also found in neurodegenerative diseases, including Alzheimer’s disease [10]. Few studies have investigated the regulation of CRIF1 and PECAM-1 in the cochlea in diabetes. This study investigated the histopathology and mechanisms in the inner ear associated with diabetes, specifically focusing on intrinsic and extrinsic apoptosis and necroptotic programmed cell death machinery.

## 2. Results

### 2.1. Body Weight and Blood Glucose Levels in Control and Diabetic Mice

The body weight and blood glucose levels are shown in Figure 1A,B. The mice were given access to food and water ad libitum. Significant differences were found in both body weight (main effect of age, F_(2,18)_ = 154.1; main effect of diabetes, F_(1,18)_ = 1482; interaction, F_(2,18)_ = 32.24; *p* < 0.0001) and blood glucose level (main effect of diabetes, F_(1,12)_ = 5346, *p* < 0.0001) at all time points between the db/+ (wild-type) control and the db/db (B6.BKS(D)-*Lepr*^db^/J) mice, confirming diabetic phenotypes, including hyperglycemia and weight gain, in an animal model of diabetes before testing hearing sensitivity.

### 2.2. Weekly ABR Threshold Measurements in Wild-Type and Diabetic Mice

Mice were subjected to weekly hearing tests from 4 weeks of age using the auditory brainstem response (ABR) to determine hearing sensitivity. The ABR threshold was significantly increased in the db/db mice as early as 6 weeks of age at 32 kHz (Figure 1F) and in all other frequencies (Figure 1C–G) at later time points, suggesting significantly decreased hearing in diabetic mice compared to the wild-type controls.

### 2.3. Hair Cell Loss and Synaptopathy in the Cochlea of Diabetic Mice

After ABR measurements were completed at 14 weeks old, the cochlea was removed for whole-mount preparations. The tissues from the WT (Figure 2(A1–A3)) and db/db (Figure 2(B1–B3)) mice were subjected to immunofluorescence staining for myosin-VIIa and neurofilament heavy chain (NF-H) to visualize the sensory hair cells and neurofilaments, respectively. We examined the survival of inner hair cells (IHCs) and outer hair cells (OHCs). As shown in Figure 2C, hair cell survival was not significantly different in the apex and middle turns of the cochlea between the db/db and WT mice, while it was significantly decreased in the base turn of the db/db compared to the WT controls (51.5 ± 1.91 HCs/100 μm vs. 58.0 ± 1.15 HCs/100 μm, respectively).

Although we observed a statistical difference in the base turn, the slight decrease in HC count was not sufficient to explain the significant reduction in the ABR threshold in the db/db mice. Therefore, we evaluated the cochlear synaptopathy associated with diabetes. The synapses between IHCs and the cochlear nerve terminals were immunostained with anti-C-terminal binding protein antibody (CtBP2, a presynaptic ribbon marker), and neurofilaments were visualized by staining with an anti-NF-H antibody. The diabetic cochlea showed a marked reduction in the number of synaptic ribbons on the IHCs (Figure 2(E1,E2)). Many more synaptic ribbons remained in all the regions of the WT cochlea (Figure 2(D1,D2)). The quantitative analyses confirmed the differences in the number of synaptic ribbons between the two groups (Figure 2F).

To further confirm the synaptopathy, we examined the ABR wave-I amplitude (Figure 3A–F), which provides a sensitive measure of auditory nerve function [14,15,16]. The ABR wave-I amplitudes were significantly lower in the db/db mice at all levels (60–90 dB SPL) at 6, 10, and 14 weeks old, consistent with the decreased number of synaptic ribbons on the IHCs, confirming significant synaptopathy in diabetic mice. These data indicate that cochlear synaptopathy contributes more to the loss of hearing sensitivity than hair cell loss does.

### 2.4. Microvasculature and Cochlear Blood Flow

To determine whether diabetes contributes to the cochlear microvasculature, platelet endothelial cell adhesion molecule (PECAM)-1 (CD 31), expressed at high levels on the surface of endothelial cells, was assessed using an immunofluorescence assay (Figure 4A,B), and cochlear blood flow (Figure 4D) was measured via laser Doppler flowmetry. A marked decrease in the PECAM-1 protein was observed in the diabetic cochlea, which was confirmed using quantitative analyses (Figure 4C). Figure 4D shows that cochlear blood flow was significantly decreased in the db/db mice compared to the WT controls, indicating impaired cochlear microcirculation in the diabetic mice.

### 2.5. Mitochondrial Damage in the Cochlea of db/db Mice

Synapses and spiral ganglion neuron (SGN) loss are important pathologies in hearing impairment [17,18,19]. The histopathological changes were focused on the mitochondria of the IHC, OHC, and synapses (Figure 5). In the TEM images, the IHCs (Figure 5A), OHCs (Figure 5B), and synapses (Figure 5C,D) in the diabetic mice showed marked mitochondrial abnormalities compared to the WT controls, including vacuolated mitochondria with disrupted cristae. Similar results were observed in the cochlear stria vascularis (Figure 6). The TEM images revealed substantial disruption of the mitochondrial structure and morphology in the stria vascularis with small vacuoles and gaps between the strial cells, specifically in the intermediate cells. The mitochondria appeared swollen and distorted with reduced cristae in the stria vascularis of the db/db mice (Figure 6B) compared to the WT controls (Figure 6A), indicating structural and morphological damage in the cochlea of the db/db mice.

### 2.6. Mitochondrial Dysfunction in the Cochlea of db/db Mice

Next, we investigated the mechanisms underlying mitochondrial damage in the diabetic mice via qRT-PCR and Western blotting analyses. As shown in Figure 7, the diabetic cochlea had significantly increased levels of the proinflammatory cytokine IL-1β, mitochondrial DNA (mtDNA), mitochondrial matrix contents (*TFAM*, *mt12stRNA*, and *mt18stRNA*), antioxidant enzymatic scavengers (superoxide dismutase (*SOD*)-2), and oxidative phosphorylation markers compared to the WT mice. The diabetic mice showed significant decreases in peroxisome proliferator-activated receptor-α coactivator 1-α (*PGC-1α*) expression, a transcriptional coactivator involved in regulating cellular energy metabolism and mitochondrial biogenesis [20,21]. Taken together, the results of the qRT-PCR indicated that a diabetic cochlea has increased levels of proinflammatory cytokines and impaired mitochondrial biogenesis and glucose and fatty acid metabolism.

To investigate mitochondrial dysfunction at the protein level, the proteins required for mitochondrial oxidative phosphorylation (cytochrome c oxidase subunit 4, COX4) [22,23] and the intramitochondrial production of mtDNA-encoded oxidative phosphorylation (OXPHOS) subunits (Crif1) [12] were measured. As shown in Figure 8, both the COX4 and CRIF1 expression levels (Figure 8A,B) were significantly decreased in the db/db mice compared to the WT controls, suggesting severely impaired mitochondrial function in the db/db mice.

### 2.7. Apoptotic Cell Death, However, Not Necroptosis, in Diabetic Cochlea

The protein levels of BCL-2, caspase-9, and caspase-8 were also assessed to examine whether diabetes induces apoptosis in the cochlea (Figure 8C). The db/db mice showed significantly decreased levels of BCL-2 and caspase-9 compared to the WT controls, indicating mitochondria-mediated apoptotic cell death in the diabetic cochlea. However, there were no differences in the expression of caspase-8, the initiator of the extrinsic apoptotic pathway, between the two groups (Figure 8C). To further investigate the cell death machinery, marker proteins of necroptosis were quantified using an ELISA. There were no differences in the expression of the necroptotic marker proteins, receptor-interacting serine/threonine-protein kinase 1 and 3 (RIPK1 and RIPK3), and the pseudokinase mixed-lineage kinase domain-like (MLKL) between the groups at 14 weeks old (Figure 8E). Our results suggest that mitochondrial-mediated apoptosis, but not necroptosis, significantly contributes to cell death in the cochlea in diabetes-induced ototoxicity.

## 3. Discussion

Ample evidence shows that the inner ear is a target for insulin resistance and insulin signaling both in humans and in animal models [24]. Our results suggest microangiopathy, mitochondrial dysfunction, and synaptopathy are responsible for hearing impairment in the inner ear of hyperglycemic, insulin-resistant mice. Unlike cochlear synaptopathy in noise- and/or age-related hearing impairment [17,18,19,25,26,27,28], that in diabetes has not been characterized in detail in humans or in animal models. Our previous study demonstrated cochlear synaptopathy in diabetic mice [29], and the results of the present study also indicate significant reductions in CtBP2 expression and synapses in the cochlea of the hyperglycemic mice compared to the WT controls.

We previously reported that the aged cochlea shows necroptotic programmed cell death in the organ of Corti and stria vascularis at both the mRNA and protein levels [30]. Aged C57BL/6J mice (20 months old) showed marked increases in the expression of all three necroptotic markers, RIPK1, RIPK3, and MLKL, in the cochlea, suggesting an association between age-related hearing loss and the activation of necroptotic cell death signaling in the inner ear. In the present study, we initially hypothesized that the hyperglycemic cochlea may undergo necroptosis in addition to apoptotic cell death during hyperglycemia-related hearing loss. Contrary to our hypothesis, the diabetic mice did not show any changes in the expression of necroptotic markers, at least in the whole cochlea of the male db/db mice at 14 weeks of age. Instead, the diabetic cochlea showed severe damage in the mitochondria, affecting both their morphology (Figure 5 and Figure 6) and function (Figure 8A), which may have later induced mitochondria-dependent apoptosis. We also found that the mitochondria of the db/db mice in the stria vascularis and organ of Corti had a swollen and distorted morphology with significantly damaged cristae. Their function was examined by analyzing the expression of COX4 and CRIF1, both of which showed significantly decreased levels in cochlear lysates. The whole-cochlear lysates (Figure 8A) showed significantly decreased CRIF1 expression levels, which may have been due, at least partially, to hyperglycemia per se and/or hyperglycemia-induced ROS generation and chronic inflammatory cytokine production. The impaired CRIF1 system was speculated to prohibit the proper function of mtDNA and the synthesis of oxidative phosphorylation complexes, thus resulting in the worsening of mitochondrial damage, where cells initiate apoptotic cell death (Figure 8C) to protect the cochlear cells from further damage by hyperglycemia-induced cytokines and other cellular stresses.

The impact of diabetes on a highly vascular organ, such as the cochlea, requires attention. PECAM-1 (CD31) is a cell-surface glycoprotein receptor expressed in vascular endothelial cells and a range of blood cells, including platelets, monocytes, B lymphocytes, some T lymphocyte subsets, and neutrophils [31]. We tested PECAM-1/CD31 activity as a surface marker expressed in the vascular endothelial cells of the stria vascularis not only because prolonged hyperglycemia adversely affects the supply of nutrients and oxygen to the microvessels of the inner ear, but also because there have been no previous studies of the association between PECAM-1 activity and diabetes/hyperglycemia in the cochlea. A previous study indicated that diabetes induced a significant decrease in PECAM-1 expression in the retina of db/db mice [32]. In vitro studies have confirmed that hyperglycemia causes PECAM-1 loss in retinal endothelial cells [33]. To the best of our knowledge, the results presented here represent the first evidence of significantly decreased PECAM-1/CD31 activity in the stria vascularis of the cochlea in diabetic mice (Figure 4A,B). It is speculated that prolonged hyperglycemia may directly and/or indirectly impact protein activity/expression in the inner ear.

In addition, PECAM-1 suppresses mitochondria-dependent apoptosis [8], where intrinsic apoptosis is mediated by mitochondrial outer membrane permeabilization (MOMP), resulting in the activation of caspase-9. We showed that prolonged hyperglycemia and its related intracellular stresses induced profound morphological and functional mitochondrial damage (Figure 5, Figure 6 and Figure 7) as well as mitochondria-dependent apoptosis through the intrinsic cell death pathway involving BCL-2 and caspase-9 activation (Figure 8). Interestingly, the extrinsic apoptosis pathway (caspase-8), initiated through transmembrane death receptors, was not activated in the cochlea of the 14-week-old db/db mice. Although further studies are still required, these data may support the important role of PECAM-1 in blocking the mitochondria-dependent apoptosis observed in hyperglycemic mice.

As mentioned above, caspase-8 is the initiator caspase of extrinsic apoptosis [34]. In addition, it plays an important role in the inhibition of necroptosis [35]. Caspase-8 deficiency causes endothelial cell necroptosis, leading to cardiovascular defects and embryonic lethality [36], which can be rescued by the deletion of either of the necroptosis mediators, Ripk3 [37] or Mlkl [38]. We found that the activator of caspase-8, TNF-α, was not increased in the cochlea of the db/db mice at 14 weeks of age (Figure 7B, mRNA level), consistent with the lack of changes in caspase-8 expression, indicating no activation of the extrinsic marker of the apoptotic pathway. Consistent with the lack of caspase-8 activation, the downstream necroptotic pathway was not activated in the db/db mice at 14 weeks of age. These data suggest that the maintenance of caspase-8 level (inactivity of extrinsic apoptotic cell death) may play a protective role in controlling the downstream cell death machineries, including necroptosis, and, therefore, the diabetic mice did not show necroptotic cell death in the cochlea (Figure 8E). A recent review also supports our hypothesis that the necroptotic cell death pathway may not significantly contribute to cell injury in diabetes [39,40] and that necroptosis may not play an essential role in cell survival in type 1 diabetes [41].

In summary, diabetes has marked impacts on mitochondria, including mitochondrial morphological damage, functional impairment (COX4 and CRIF1), and mitochondria-mediated intrinsic apoptosis, but neither extrinsic apoptosis nor necroptosis, at least at 14 weeks of age in the cochlea of db/db mice. We speculated that older diabetic mice may show the activation of caspase-8 and necroptosis mediators, and future studies may refine our conclusions.

## 4. Materials and Methods

### 4.1. Research Design and Experimental Animals

Male B6.BKS(D)-*Lepr*^db^/J (db/db, diabetes, 22 mice) and heterozygote (db/+, control, 20 mice) mice were purchased (total of 42 mice) from Jackson Laboratory (Bar Harbor, ME, USA). Animals within each group (control, diabetes) were randomly assigned to experiments (random allocation), and a single animal was considered as an experimental unit except for molecular tests (single cochlea/unit). There were no exclusions of the aniμal during analysis. All mice were maintained in an identical environment (temperature 22 °C, humidity 45–55%) with a 12/12-h dark–light cycle (0700 to 1900 h) and fed pelleted food (Envigo, #2018C Teklad Global 18% Protein Rodent Diet) and water ad libitum. All animal procedures were approved by the IACUC of Chungnam National University (2019012A-CNU-169, 15 December 2019).

### 4.2. Data and Resource Availability

The datasets of the current study are available from the corresponding authors on reasonable request.

### 4.3. Auditory Brainstem Response

ABR thresholds at frequencies between 4 and 32 kHz and click sounds were obtained separately from both ears, as described previously [42]. The TDT System-3 (Tucker Davis Technologies, Gainesville, FL, USA) hardware and software were used to obtain the ABRs. The stimuli were computer-generated tone pips. The animals were anesthetized with intramuscular injection of zolazepam HCl 40 mg/kg (Zoletil, Virbac Animal Health, Carros, France) and xylazine 10 mg/kg (Rompun, Bayer Animal Health, Monheim, Germany). Subcutaneous needle electrodes were placed around the skull vertex and both infraauricular areas. Tone bursts, with a duration of 4 ms and rise-fall time of 1 ms at frequencies of 4, 8, 16, and 32 kHz, were used, in addition to clicks. The sound intensity was varied in 5 dB increments for the tone burst sounds and the click. The contralateral ear was not masked because the stimuli were transmitted through a sealed earphone. The waveforms were analyzed using a custom program (BioSig RP, ver. 4.4.1; Tucker Davis Technologies) with the researcher blinded to the treatment group. ABR threshold was defined as the lowest stimulus intensity to evoke a wave III response > 0.2 μV. The amplitude of short latency responses (wave 1) was recorded to assess auditory nerve survival/synaptopathy as previously reported [43].

### 4.4. Immunostaining

Cochlear tissues were fixed in 4% paraformaldehyde in phosphate-buffered saline (PBS) for 1 h at room temperature. After removal of the cochlear bony walls and lateral wall tissues, the remaining cochlear tissues were prepared for immunostaining. Tissues were permeated with 0.3% Triton X-100 (Sigma-Aldrich, St. Louis, MO, USA) for 10 min, blocked in 5% normal goat serum (Vector Laboratories, Burlingame, CA, USA) for 30 min, and then incubated with anti-myosin VIIa primary antibody (Proteus BioSciences, Ramona, CA, USA), -neurofilament heavy chain (NF-H) primary antibody (NOVUS, #NBP197726), -CtBP2 primary antibody (BD bioscience, #612044), or -Hoechst 33342 primary antibody (Life technologies, #H3570) at a concentration of 1:200 in blocking solution overnight at 4 °C. After rinsing in PBS for 10 min (3×), the tissues were incubated with the AlexaFluor 594 (Invitrogen, #A11037) or the AlexaFluor 488 (Invitrogen, #A11039) at a concentration of 1:200 in PBS for 2 h. After rinsing in PBS for 10 min (3×), specimens were mounted on glass slides using Crystalmount (Biomeda, Foster City, CA, USA). The specimens were observed under an epifluorescence microscope (Zeiss Axio Scope A1; Zeiss, Oberkochen, Germany) with a digital camera.

### 4.5. Transmission Electron Microscope (TEM)

Tissue samples were fixed with 3% glutaraldehyde in culture medium for 2 h at room temperature. They were washed five times with 0.1 M cacodylate buffer containing 0.1% CaCl_2_ at 4 °C. Then, they were postfixed with 1% OsO_4_ in 0.1 M cacodylate buffer (pH 7.2) containing 0.1% CaCl_2_ for 2 h at 4 °C. After rinsing with cold distilled water, tissue samples were dehydrated slowly with an ethanol series and propylene oxide at 4 °C. The cells were embedded in Spurr’s epoxy resin [44]. After polymerization of the resin at 70 °C for 36 h, serial sections were cut with a diamond knife on an ULTRACUT UCT ultramicrotome (Leica Mikrosysteme GmbH, Vienna, Austria) and mounted on formvar-coated slot grids. Sections were stained with 4% uranyl acetate for 10 min and lead citrate for 7 min [45]. They were observed using a Tecnai G2 Spirit Twin transmission electron microscope (FEI Company, Hillsboro, OR, USA).

### 4.6. Measurement of Cochlear Blood Flow

The left tympanic bulla of each mouse was exposed and opened under anesthesia. After the mouse was placed on a heating pad, the cochlear blood flow was measured using a 0.1-millimeter-diameter laser Doppler probe placed over the lateral wall of the cochlea. Cochlear blood flow was determined from an intensity oscillation that was translated from the frequency of the oscillation produced by the Doppler frequency shift of the red blood cells in the left tympanic bulla, using a Laser Doppler Flowmeter (Transonic Systems, Ithaca, NY, USA). Each intensity oscillation was measured separately, and relative cochlear blood flow was reported as the ratio of the control (db/+) value to the value of diabetic (db/db) mice.

### 4.7. Quantitative Real Time Polymerase Chain Reaction (qRT-PCR)

Quantitative RT-PCR was performed as previously described [29,30,42]. Briefly, tissues were collected and frozen immediately in liquid nitrogen and homogenized. Total RNA was extracted with TRIzol reagent (Thermo Fisher Scientific, Waltham, MA, USA) according to the manufacturer’s protocol. RNA was quantified using a Nanodrop (Nanodrop Technologies, Wilmington, DE, USA). cDNA was produced using the cDNA synthesis kit (Roche, Branchburg, NJ, USA). Real time PCR was performed on a CFX Connect Real-Time PCR Detection System (BioRad, Des Plaines, IL, USA) by using a reaction mixture with SYBR Green as the fluorescent dye (Applied Biosystems, Waltham, MA, USA), a 1/10 vol of the cDNA preparation as template, and 250 nM of each primer (Realtime primers, PA, USA). The fold change in the target gene relative to endogenous control gene was determined by the following: fold change = 2^−Δ(ΔC^_T_^)^, where ΔC_T_ = C_T, target_ − C_T, 18S_ and Δ(ΔC_T_) = ΔC_T, Diabetes(db/db)_ − ΔC_T, Control(db/+)_.

### 4.8. Western Blotting

Tissues were collected and lysed in RIPA buffer (Sigma, St. Louis, MO, USA) in the presence of protease and phosphatase inhibitors (Halt™ protease and phosphatase inhibitor, Thermo Scientific, Waltham, MA, USA), and samples were prepared in protein sample buffer (0.25% bromophenol blue, 0.5M dithiothreitol, 50% glycerol, 10% sodium dodecyl sulfate (SDS), 0.25M Tris-Cl pH 6.8, and trace amounts of bromophenol blue), boiled, and stored at −80 °C. For gel electrophoresis and Western blot analyses, samples were run on 4–15% precast Tris-HCl SDS-polyacrylamide gels (BioRad, Hercules, CA, USA) and transferred to polyvinylidenedifluoride (PVDF) membranes (Millipore, Burlington, MA, USA). Blots were successively probed with anti-COX4 (Cell signaling technology, #11967S), -CRIF1 (Santa Cruz, #sc-374122), -caspase 8 (Cell signaling technology, #4790S), -caspase 9 (Cell signaling technology, #9508S), -BCL-2 (Cell signaling technology, #2876S), or -β-actin (Cell signaling technology, #4967S) antibodies at 1:1000 dilutions in TBS containing 3% protease free BSA (Sigma, St. Louis, MO, USA). Blots were visualized using Immobilon Western Chemiluminescent HRP Substrate (Millipore, Burlington, MA, USA) and the images were acquired and quantitated using Azure 300 Chemiluminescent Western Blot Imaging System (Azure Biosystems, Dublin, CA, USA).

### 4.9. Measurement of Tissue Levels of RIPK1, RIPK3 and MLKL

Cochlear necroptosis markers were measured using standard sandwich enzyme-linked immuno-sorbent assay mouse RIPK1 (Aviva Systems Biology, San Diego, CA, USA, #OKCA01762), RIPK3 (Aviva Systems Biology, San Diego, CA, USA, #OKCD02851), and MLKL (Aviva Systems Biology, San Diego, CA, USA, #OKEH05350) ELISA Kits as per the manufacturer’s instruction. Samples, standards, and controls were added to appropriate wells in a 96-well plate, as stated in the kit protocol, and incubated at room temperature for 2 h on a horizontal orbital microplate shaker. After washing, 50 μL of conjugate was added to each well and incubated at room temperature for an hour on the shaker. After washing and incubation in substrate solution for 30 min, the plates were read at 450 nm in a plate reader (Tecan US Inc., Durham, NC, USA). Sample measurements were interpolated from the standard curve, and values from tissue lysates were normalized to total protein concentrations.

### 4.10. Image Processing and Statistical Analysis

Adobe Photoshop CS6 was used for the adjustment of image contrast, superimposition of images, and colorization of monochrome fluorescence images. An unpaired Student’s *t*-test or a one-way ANOVA with Bonferroni’s multiple comparisons test were used for comparisons. A *p*-value < 0.05 was significant in each case. All tests were performed using GraphPad Prism 6 (GraphPad Software, San Diego, CA, USA). All measurements were taken from distinct samples and sample sizes were determined without any expectation of the effect size. All experiments were repeated multiple times, and the exact number of sample size allocated to each group was described within the main body of the text and/or in the figure legends. Hearing tests and all molecular tests were performed in a blinded manner.

## Figures and Tables

**Figure 1 ijms-22-08807-f001:**
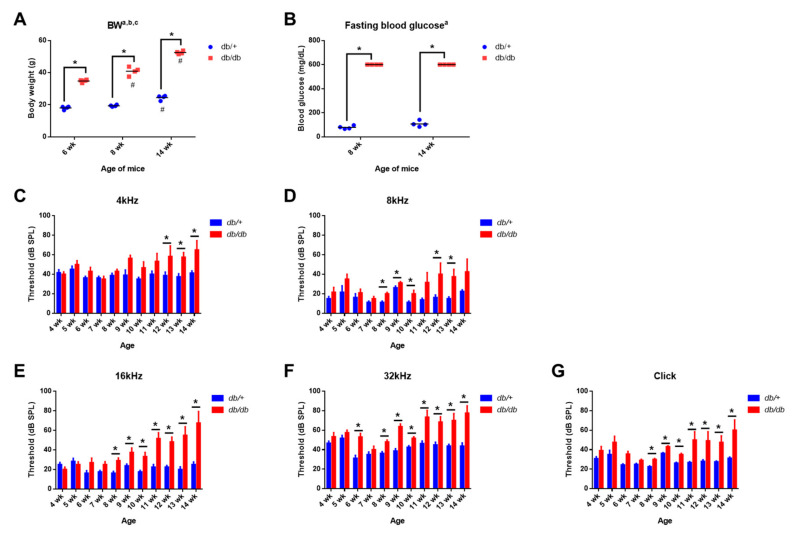
Diabetic (db/db) mice show hyperglycemia and impaired hearing. (**A**,**B**) Increased body weight and hyperglycemia developed in male B6.BKS(D)-*Lepr*^db^/J mice. Significant differences were found in body weight (**A**) and blood glucose levels (**B**) between db/db and db/+ mice at all time points. Maximum reading from the glucometer was recorded as 600 mg/dL. *n* = 4 per group. All graphs represent mean ± S.E.M. Asterisk (*) denotes *p* < 0.05 and pound symbols (#) denote differences from matching 6 wk values; a, main effect of diabetes, *p* < 0.0001; b, main effect of time, *p* < 0.0001; c, interaction, *p* < 0.0001, two-way ANOVA, Tukey’s multiple comparisons test. (**C**–**G**) Diabetic (db/db) mice displayed a decreased hearing sensitivity compared to wild type animals. Auditory brainstem response (ABR) thresholds from db/+ and db/db mice were recorded every week between 4- and 14-weeks of age at 4 (**C**), 8 (**D**), 16 (**E**), and 32 (**F**) kHz and click (**G**) sound. ABR thresholds were significantly increased in the db/db animals compared to the db/+ mice. *n* = 6–10. All graphs represent mean ± S.E.M. * *p* < 0.05. Unpaired *t*-test.

**Figure 2 ijms-22-08807-f002:**
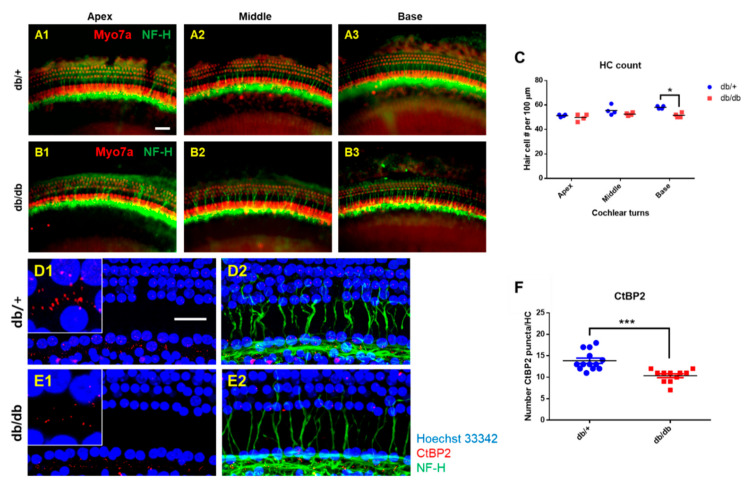
Hair cell loss and synaptopathy in the cochlea of diabetic mice. (**A**–**C**) Whole mounts of the auditory epithelium from db/+ and db/db at 14 weeks of age. Tissues were visualized by myosin VIIa (red, hair cells) and NF-H (green, neurofilaments), and photographed using a fluorescence microscope (**A**,**B**). Hair cells from apex and middle turns of the cochlea did not show differences between db/+ and db/db mice (**C**). A significant hair cell loss was found in the base turn of db/db mice compared to db/+ animals (**C**). *n* = 4. (**D**–**F**) Loss of nerve fibers and synaptic ribbons in diabetic mice at 14 weeks of age. (**D**,**E**) Whole-mounts of auditory epithelium were triple-stained with CtBP2 (red, a marker of synaptic ribbons), NF-H (green, neuronal cell marker), and Hoechst (blue, nuclear marker) to evaluate synaptopathy. Severe synaptic loss ((**D1**) vs. (**E1**), inset, red) and decreased nerve fibers (green) were observed in the db/db mice compared to the db/+. Number of pre-synaptic marker (CtBP2, red) per 10 hair cells (blue) was quantified (**F**). Diabetic mice (db/db mice) showed a significant decrease in CtBP2 counts. *n* = 11. * *p* < 0.05, *** *p* < 0.001. Scale bar = 30 μm. All graphs represent mean ± S.E.M. Unpaired *t*-test.

**Figure 3 ijms-22-08807-f003:**
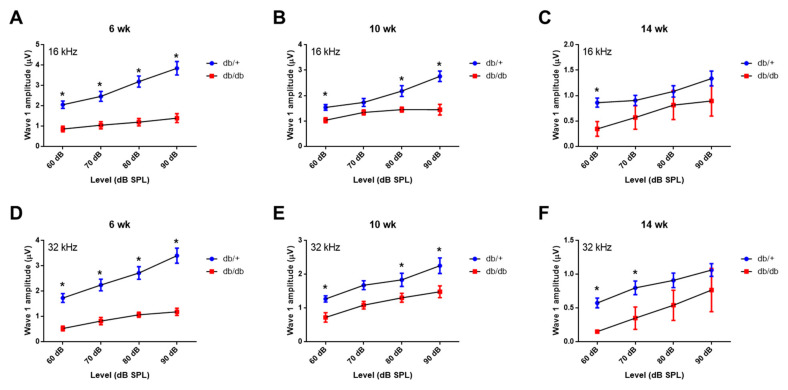
Auditory nerve function was significantly reduced in diabetic mice compared to wild type. ABR wave I amplitudes (a sensitive measure of auditory nerve function) were elicited by tone pips at 16 (**A**–**C**) or 32 kHz (**D**–**F**) at 6, 10, and 14 weeks of age. *n* = 10. Asterisk indicates *p* < 0.05, unpaired *t*-test.

**Figure 4 ijms-22-08807-f004:**
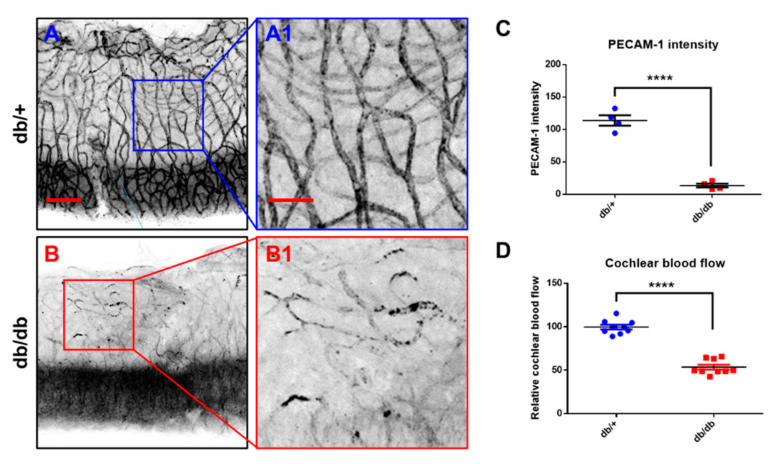
Changes of cochlear blood flow and microvasculature of the stria vascularis. (**A**,**B**) Immunofluorescence measurements of platelet endothelial cell adhesion molecule-1 (PECAM-1/CD31) in the vessels of the stria vascularis. Endothelial cells of stria vascularis were stained with anti-PECAM antibody and imaged with a confocal microscope. Scale bar represents 100 µm (**A**,**B**) and 40 µm (**A1**,**B1**). (**C**) Intensity of PECAM-1/CD31 positive area was quantified. PECAM-1/CD31 staining was significantly reduced in db/db mice in the vessels of the stria vascularis. *n* = 4. (**D**) Cochlear blood flow was significantly decreased in the db/db mice compared to the db/+ animals. *n* = 9. **** *p* < 0.0001. Unpaired *t*-test.

**Figure 5 ijms-22-08807-f005:**
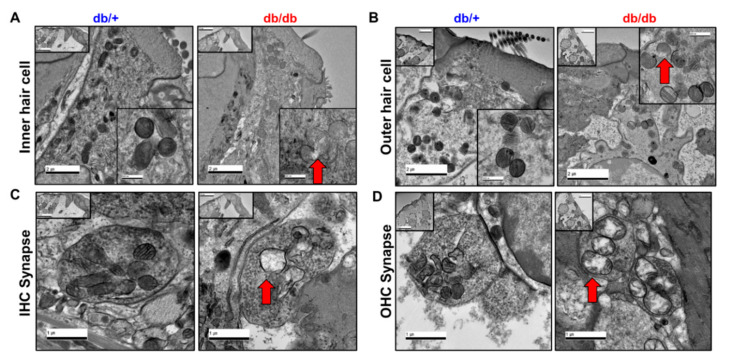
Representative transmission electron microscopy (TEM) images of mitochondria from cochlear IHC, OHC, and synapses. Cochlea mitochondria of diabetic mice were significantly damaged in the inner (**A**) and outer hair cells (**B**) and synapses (**C**,**D**) compared to wild-type. Mitochondria from diabetic cochlea contain vacuolated mitochondria with disrupted cristae. Scale bars, (**A**,**B**), 2 μm, insert, 500 nm; (**C**,**D**), 1 μm, insert, 20 μm.

**Figure 6 ijms-22-08807-f006:**
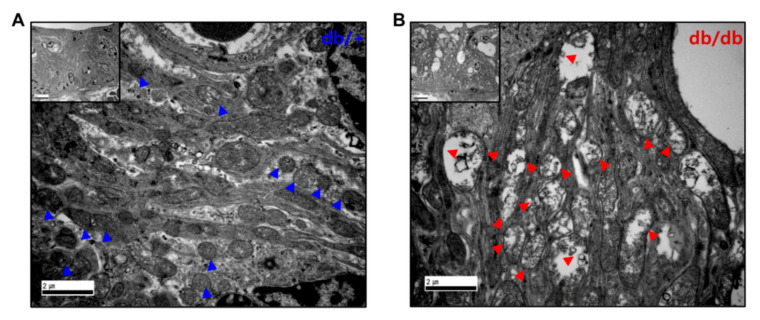
TEM images of the cochlear stria vascularis (SV). (**B**) Stria vascularis of db/db mice exerted small vacuolization and gaps between the strial cells. Mitochondria appeared swollen and distorted with reduced cristae in the SV of db/db mice (**B**), red triangles compared to db/+ animals (**A**), blue triangles. Scale bar = 2 μm, insert, 5 μm.

**Figure 7 ijms-22-08807-f007:**
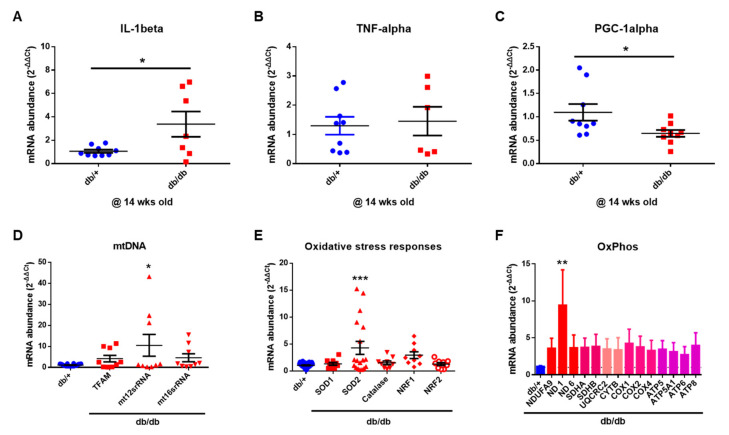
Expression of inflammatory cytokines and mitochondrial function. (**A**–**C**) Inflammatory cytokine expression in the cochlea. *n* = 9 (db/+) and *n* = 6–9 (db/db). (**D**–**F**) Mitochondrial oxidative phosphorylation and oxidative stress markers in the cochlear. * *p* < 0.05; ** *p* < 0.01; *** *p* < 0.001. Unpaired *t*-test (**A**–**C**) or one-way ANOVA (**D**–**F**) with Bonferroni’s multiple comparisons test.

**Figure 8 ijms-22-08807-f008:**
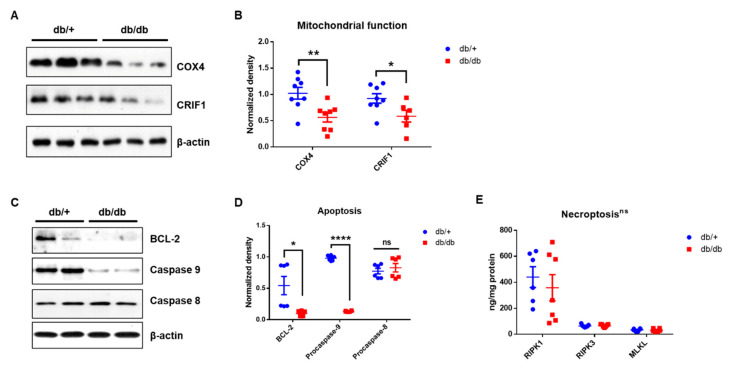
Dysfunctional mitochondria in the cochlea of diabetic mice. (**A**,**B**) Proteins required for mitochondrial oxidative phosphorylation (Cytochrome c oxidase subunit 4, COX4) and the intramitochondrial production of mtDNA–encoded OXPHOS subunits (CRIF1) were measured using Western blot. Both proteins were significantly reduced in db/db mice compared to db/+ animals. ** *p* < 0.01, * *p* < 0.05, unpaired *t*-test. *n* = 8. (**C**,**D**) Diabetic cochlea displayed an activation of BCL-2 and caspase-9, but not caspase 8. **** *p* < 0.0001, * *p* < 0.05, unpaired *t*-test. *n* = 6. (**E**) Necroptosis markers, RIPK1, RIPK3, and MLKL, showed no significant changes between db/db (*n* = 7) and db/+ (*n* = 6) mice. ns, not significant.

## Data Availability

The datasets generated during and/or analyzed during the current study are available from the corresponding author on reasonable request.

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
