# Peer review of "Hearing Impairment in a Mouse Model of Diabetes Is Associated with Mitochondrial Dysfunction, Synaptopathy, and Activation of the Intrinsic Apoptosis Pathway"

_ijms, 2021, doi:10.3390/ijms22168807_

Round 1

Reviewer 1 Report

In this manuscript, the authors studied hearing impairment in diabetic mouse model. They found that mitochondrial dysfunction, synaptopathy and activation of intrinsic apoptosis pathway are associated with this condition. This manuscript is overall interesting, generally scientifically sound and clearly written. I only have a very minor suggestion for the authors. There are various typo mistakes in this manuscript. To name a few, line 16 "inreased" should be "increased", line 31 "itohondria" should be "mitochondria", etc. Therefore, a careful text editing and proofreading by a native English speaker is highly recommended.

Author Response

We appreciate the careful reading and comments. We made changes to the manuscript (line #16 and 31). Thank you very much for catching this error.

Reviewer 2 Report

The manuscript by lyu et al entitled “Hearing Impairment in a Mouse Model of Diabetes is Associated with Mitochondrial Dysfunction, Synaptopathy and Activation of the Intrinsic Apoptosis Pathway” describes studies on hearing impairment induced due to high blood glucose level in diabetic mice. The authors have used diabetic homozygous, heterozygous and normal control mice. They studied audiometric brain response and also measured cochlear blood flow and cochlear necroptosis markers. Their results led to the conclusion that diabetes induces damage of cochlear hair cells and degenerate nerve fibers.

The findings reported in the manuscript are of interest to the readers of this journal. However, the manuscript needs improvements as given in the below comments:

Suggestions/Comments:

  1. Authors need to also give data on control mice (Figure 1-8) to conclude a decrease in hearing ability.
  2. On page 6, line 243 and 244; it is reported that “inner and outer hair cell survival in cochlea is different in diabetic and control mice. However, no fluorescent images are given of control mice.
  3. Have authors tried to reverse the hearing loss in diabetic mice if given the healthy / controlled diet. It will be worth checking the effect on nerve fibers if the blood glucose level remains under control. These results will be important for therapy in diabetic patients.

This reviewer recommends acceptance with the above provisos.

Author Response

  1. Authors need to also give data on control mice (Figure 1-8) to conclude a decrease in hearing ability.
  2. On page 6, line 243 and 244; it is reported that “inner and outer hair cell survival in cochlea is different in diabetic and control mice. However, no fluorescent images are given of control mice.

We appreciate the careful reading and comments provided by the reviewer.

Heterozygous (db/+) mice served as the control group throughout the study (Figure 1-8) because heterozygotes has exactly the same genetic background with homozygotes (B6.BKS(D)-Leprdb/J mice = db/db) except one of the alleles encoding the leptin receptor. Therefore, we used the term “control” and “db/+” interchangeably. We regret this may have caused confusion.

As a reference, B6.BKS(D)-Leprdb/J mice is the db/db mouse model of leptin deficiency (homozygous=db/db), which is the most widely used/accepted mouse model of type 2 diabetes.

  1. Have authors tried to reverse the hearing loss in diabetic mice if given the healthy / controlled diet. It will be worth checking the effect on nerve fibers if the blood glucose level remains under control. These results will be important for therapy in diabetic patients.

We have investigated the impact of diabetes on the inner ear, however, we did not test the methods to reverse diabetes-related hearing impairment in the current animal model. This is an excellent suggestion by the reviewer, and we would love to explore a therapy to rescue hearing loss induced by metabolic diseases in the future studies.

Round 2

Reviewer 2 Report

Authors have duly incorporated acknowledged all the comments and suggestions of this reviewer. As a consequence the revised manuscript looks considerably improved.

Recommended for acceptance.